# Analysis of the Quality of Sulfomolybdenum Coatings Obtained by Electrospark Alloying Methods

**DOI:** 10.3390/ma14216332

**Published:** 2021-10-23

**Authors:** Oksana P. Gaponova, Bogdan Antoszewski, Viacheslav B. Tarelnyk, Piotr Kurp, Oleksandr M. Myslyvchenko, Nataliia V. Tarelnyk

**Affiliations:** 1Applied Material Science and Technology of Constructional Materials Department, Sumy State University, 40007 Sumy, Ukraine; 2Laser Research Centre, Faculty of Mechatronics and Mechanical Engineering, Kielce University of Technology, Al. Tysiąclecia P.P. 7, 25-314 Kielce, Poland; b.antoszewski@tu.kielce.pl; 3Technical Service Department, Sumy National Agrarian University, H. Kondratiieva Str. 160, 40021 Sumy, Ukraine; tarelnyk@ukr.net; 4Department of Physical Chemistry of Inorganic Materials, Frantsevich Institute for Problems of Materials Science, Krzhizhanovsky Str. 3, 03142 Kyiv, Ukraine; zvyagina47@gmail.com; 5Projection of Technical System Department, Sumy National Agrarian University, H. Kondratiieva Str. 160, 40021 Sumy, Ukraine; natasha-tarelnik@ukr.net

**Keywords:** electrospark alloying, sulfomolybdenum coatings, tribology

## Abstract

The authors of this paper have attempted to improve the quality of surface layers applied to steel elements of machine parts constituting friction couples. The main goal of the research was to investigate an electrospark alloying method process for obtaining abrasion-resistant tribological coatings containing molybdenum disulfide on a steel surface. A substance in the form of sulfur ointment with a sulfur content of 33.3% was applied on the surfaces of C22 and C40 steel specimens. In order to determine the influence of the energy parameters of ESA equipment on the quality parameters of coatings, the ESA process was carried out using a molybdenum electrode with discharge energies Wp = 0.13; Wp = 0.55; Wp = 3.4 J. The following tests were carried out on specimens with such coatings: metallographic analysis, microhardness tests, surface roughness, and local X-ray diffraction microanalysis. The experiments revealed that sulfomolybdenum coatings consist of four zones with different mechanical properties. Depending on the discharge energy and the substrate material, the hardness of these zones varies from approx. 1100 to over 10,000 MPa. Differences in the distribution of, among others, sulfur and molybdenum in the obtained coatings, as well as differences in the microstructure of the observed coatings, were observed.

## 1. Introduction

Wear of contact joint parts, in particular of antifriction purpose, is a cause of unbalance of machine units due to changes in the sizes of the worn parts thereof, which results in equipment operation instability, loss of productivity, and reduced product quality [1,2]. The authors [3,4] have proven that the amount and term of overhaul work depend on friction connections in 70% of cases. Reducing friction and, therefore, wear of metal surfaces in the places of interactions of main component parts of machines and mechanisms is an urgent problem of modern science and technology.

There is a solid lubricant, namely molybdenum disulfide MoS_2_ [5,6], which is chemically and thermally stable up to 600 °C, and this property guarantees the above solid lubricant presence in its unchanged form in a composite material both while manufacturing and also operating. Just molybdenum disulfide is an effective lubricant in the air under light-duty or reasonable operating conditions of parts [7].

It is well known that to increase the wear resistance of machine parts, oils and lubricants laced with antifriction additives are used. One such antifriction additive is molybdenum disulfide (MoS_2_). Hexagonal MoS_2_ crystals have a layered structure. The Mohs scale molybdenum disulfide hardness is 1.0–1.5. It has a high chemical stability and resistance to most acids and radioactive irradiation. In a vacuum, molybdenum disulfide (MoS_2_) decomposes into molybdenum and sulfur at 1100 °C [8].

The lubricating properties of molybdenum disulfide are provided by its crystalline structure: Van der Waals bonds between the sulfur layers allow the latter to freely move, which results in reduced friction. On the other hand, the ionic bonds between Mo and S impart high strength to the layers, so they are able to resist punching performed by the microprotrusions of the friction surfaces. The 2.5-μm-thick lubricating layer contains 4000 layers of (S-Mo-S) (see Figure 1). The sulfur layers, which form the surface of a molybdenum disulfide crystal, provide strong adhesion to a metal surface [9,10].

At present, micron-sized powders of naturally layered molybdenum disulfide are widely used as solid lubricants and effective additives to oils and lubricating substances in order to improve their tribological properties. Advances in nanotechnology have stimulated the development of research on the processes and mechanisms of wear with the participation of nanocrystalline compounds, as well as comparisons of traditional concepts of lubrication and “nano-lubrication” [11,12]. When using solid lubricants, namely molybdenum trisulfide (Mo_3_) nanoparticles, there appears to be a positive effect on the improvement of the tribological properties of lubricants [13,14,15].

The interaction of Mo molybdenum with S sulfur is the subject of a long-term investigation. As follows from Figure 1, which shows the phase diagram of the Mo-S system at atmospheric pressure, where G is a gas and L is a liquid solution [16], in the Mo-S system, two stable molybdenum sulfides (Mo_2_S_3_ and MoS_2_) are formed, which can be directly obtained from the elements. The Mo_2_S_3_ phase is not stable at temperatures below 610 °C. In the phase of MoS_2_, up to 69% (at.) of S sulfur can be dissolved at 1000 °C. The melting point of MoS_2_ is above 1800 °C. The solubility of S in Mo reaches 1.5% (at.) at 1100 °C.

There has been prior research on thesulfo-cyanidation of electrodeposited Fe-Mo coatings [17]. The authors experimentally set the most rational sulfo-cyaniding medium, namely, a pasty coating for parts, consisting of 35–40% potassium hexacyanoferrate (II), 15–20% sulfur pyrite, 5–8% fire-clay grog, and 25–30% soot with a binder (i.e., starch glue). The process of sulfo-cyanidation is carried out at temperatures of 550 to 600 °C, which allows for a wear-resistant coating with a surface carbonitrided zone saturated with iron sulfides and molybdenum disulfide to be obtained. However, this technology is environmentally unsafe.

In [18], a new technology for forming a package of flexible elements for an elastic coupling was developed. The additional incorporation of molybdenum disulfide into the metal-cladding lubricant, which consists of paraffin and copper powder, allows for the fretting resistance of the flexible elements to be increased by ~2.5 and 1.1 times, respectively, as compared with the flexible elements without any lubricant and those with a metal-cladding lubricant without molybdenum disulfide.

In [19], one of the ways to increase the level of the operational reliability of diesel fuel equipment, namely the plunger pairs of the fuel pumps, is presented. A technological approach in restoring the efficiency of parts by applying thereon an antifriction coating containing molybdenum disulfide with the use of a vacuum-plasma spraying method is presented. Applying this technology allows the operational properties of the plunger pair component parts to be increased, as well as their resource.

Furthermore, there are known methods for chemical heat treatment (CHT), which, at sequentially or simultaneously saturating steel surfaces with Mo and S, allow sulfomolybdenum coatings to be obtained [20,21].

Despite the fact that the CHT method significantly improves the quality of the machine parts’ surface layers, the above method has a number of disadvantages:there is volumetric heating of the parts, resulting in the occurrence of changes in their structures and initial geometric parameters (deformations and curvatures);there is a need for bulky and expensive technological equipment;there is a long process duration;there is high energy consumption;there are issues regarding marginal cost;there are environmental- and technogenic-related concerns, etc.

There is a method of electrospark alloying (ESA) that is becoming more widely used in industry to increase the wear resistance and hardness of the surfaces of machine component parts, including those operating in conditions of elevated temperatures and aggressive environments [22,23].

The surface ESA is a process of transferring a curtain material to the treated surface by spark electric discharge. The method has a number of specific features:An anode material (an alloying one) can form a coating layer on the surface of the cathode (the surface being alloyed), with the coating layer being extremely tightly adhered to the above surface. In this case, not only is there an absence of an interface between the deposited material and the base metal, but there is also an even diffusion of the anode elements into the cathode;The alloying process can be carried out in specified places (with the radius range starting from particles the size of a millimeter and larger) without protecting the rest of the surface of the part being alloyed;The application ofthe ESAmethod on metal surfaces is very simple, and the necessary equipment is compact and transportable [24].

The ESA method expands the possibilities of obtaining tribological coatings. In [25], a process for applying a wear-resistant antifriction coating on the working surface of a piston ring is described. It consists of obtaining a coating of molybdenum of a thickness of 30 μm, microhardness of 6400 MPa, and roughness of Rz 1.5 to 1.1 μm. The alloyed surface is subjected to manual sanding with sandpaper and coating with copper. Furthermore, the surface obtained is treated by the method of surface plastic deformation followed by the application of a consistent lubricant, which contains molybdenum disulfide and graphite, to improve the friction unit’s running-in ability. However, this technology has significant drawbacks, including an increase in the number of technological operations, the presence of the lubricant on the surfaces being treated, and the risk of contamination of the working environment of the part, which is extremely undesirable, for example, in the friction units of equipment for the textile and food industries, etc.

There is a process of sulfocarburizing a steel product surface by the method of electroerosion alloying (EEA) with the use of a graphite electrode, wherein immediately before alloying with the graphite electrode, a grease lubricant containing sulfur is applied onto the steel surface [26]. The main disadvantages of this method are as follows:there is an inability to form molybdenum disulfide on the surface of the parts;there is a decreased possibility of reducing the coefficient of friction;there is insufficiently high running-in and wear resistance within the friction pairs, especially without lubrication.

We have proposed a method for sulfomolybdenum metal surfaces, which include materials coated with a substance in the form of a sulfur ointment and electrospark alloying with a molybdenum electrode. In our opinion, this technology will make it possible to obtain self-lubricating coatings, as in [26], and to provide special tribotechnical properties (for example, a low coefficient of friction).

Thus, the aim of the present work is to improve the quality of the surface layers of steel parts involved in friction pairs. The problem has been solved by developing a process for obtaining wear-resistant tribological coatings, which contain molybdenum disulfide, on the steel surfaces by applying the method of electrospark alloying.

## 2. Materials and Methods

To determine the influence of the energy parameters of ESA equipment on the quality parameters of coatings, specimens sized 15 × 15 × 8 mm and made of C22 steel and C40 steel were coated with a substance in the form of a sulfur ointment with a sulfur content of 33.3%. After that, the ESA process was performed using a molybdenum electrode of 4 mm diameter and 45 mm length using the Elitron—52A unit under the ESA operating modes at discharge energy Wp = 0.13; 0.55; 3.4 J. Each ESA operating mode corresponded to its discharge energy and productivity, namely, the area of the formed coating per time unit (see Table 1).

The metallographic analysis of the coatings was carried out using the MIM-7 optical microscope (LOMO, Saint Petersburg, USSR); microhardness testing was performed on the PMT-3 instrument (LOMO, Saint Petersburg, USSR) according to standard methods. The surface roughness after ESA was determined by reading and processing the profilograms using the profilograph-profilometer device (Kalibr Instrument Plant, Moscow, Russian Federation) of model 201 manufactured at the “Caliber” plant.

To study the layer depth distribution of the elements, a local X-ray diffraction microanalysis was performed based on the registration of the characteristic X-rays excited by an electron beam of the chemical elements that were available in the microvolume. To this end, a scanning electron microscope of the SEO-SEM Inspect S50-B (FEI Company, Oregon, USA) type equipped with the energy-dispersive spectrometer of the AZtecOne model with the detector of the X-MaxN20 type was applied (Oxford Instruments plc, Abingdon-on-Thames, Great Britain). We used the following parameters: working distance (WD) 10.2 mm, spot size (spot) 6.0, accelerating voltage (HV) 20.00 kV.

The X-ray studies were performed in CoKα-radiation with the use of the diffractometer of the DRON-UM1 model (LNPO “Burevestnik”, St. Petersburg, Russian Federation). The diffractograms were taken by the step-by-step scanning method. The scanning step was 0.050; the exposure time at a point was 3 s. The processing of the diffractograms was performed using a program for the full-scale analysis of X-ray spectra from the mixture of the polycrystalline components of the Powder Cell 2.4 program.

## 3. Results and Discussion

The metallographic studies and microhardness testing of the coatings showed that the sulfomolybdenum coatings consist of four zones: the upper loose layer of microhardness Hμ = 1112 ÷ 2040 MPa, a strengthened layer called “white” of Hμ = 5147 ÷ 5474 MPa for Wp = 0.13 J, as well as of Hμ = 10596 ÷ 10731 MPa for Wp = 3.4 J, the diffusion zone, and the base metal (see Figure 2, Figure 3, Figure 4 and Figure 5).Upon replacing the substrate of C22 steel with the substrate of C40 steel, an increase in the microhardness and the thickness of the strengthened layer occurs, as well as the continuity thereof (Table 2).

A scanning electron microscope with a microanalysis system was used to study the sulfomolybdenum coatings. The image obtained was examined with the BSE detector of backscattered (reflected) electrons. The above image contains the well-visible and clearly defined areas, which differ in tone color depending on the atomic number of the chemical element. In such images, the light regions are the areas containing the heavier elements (in our case, molybdenum), from which the electrons of the beam are reflected better than from the lighter elements.

The results of the analysis of the surface area, which includes the coating and the fragment of the base metal with the pre-eutectoid structure, showed that the resulting layer has a heterogeneous composition with different concentrations of elements (see Figure 6). Thus, according to the element distribution maps of the areas of the studied specimens (see Figure 7), sulfur is concentrated on the surface and molybdenum is more evenly distributed in the coating.

As a result of the local energy dispersion X-ray microanalysis, the distribution curves of sulfur, molybdenum, and iron were obtained (see Figure 8). It was observed that sulfur and molybdenum were concentrated at depths of up to 4 and 19 μm, respectively, at Wp = 0.13 J; up to 5 and 25 μm, respectively, at Wp = 0.55 J; and up to 15 and 70 μm, respectively, at Wp = 3.4 J.

The X-ray diffraction analysis of the obtained coatings confirmed the results of the energy dispersion analysis (see Figure 9). Thus, at low discharge energies, the phase composition of the coatings on C40 steel was represented by the BCC (body centered cubic) solid solution with a lattice period close to ferrite, martensite, the FCC (face centered cubic) solid solution, and the FeMo intermetallide (σ-phase). Obviously, because of the alloying of the BCC solid solution (ferrite) by sulfur and molybdenum, as well as the processing under the non-equilibrium cooling conditions, the parameter *a* increased there (see Table 3).

Under conditions of high heating and cooling rates of the surface layer microvolumes, which lead to the formation of nonequilibrium structures, and also as a result of mixing the base material and the material of the alloying electrode, due to the interaction between the liquid alloy bath and the environment, namely air, during the ESA process, as well as owing to the intense shock waves that appear during the ESA process and lead to thermo-mechanical hardening, and other events that influence the phase formation, it is obvious that two alloyed austenites were formed in the surface layer at alloying temperatures greater than 1000 °C. One of them, the Mn martensitic point, which is above room temperature, when cooled, undergoes martensitic transformation, thus forming martensite with lattice parameters *a* = 2.8740 nm and *c* = 2.9200 nm. Due to the fact that molybdenum intensively reduces the temperature of martensitic transformation, it is not going to end, and in the coating, there remains the unconverted residual austenite, namely, the FCC solid solution (see Table 3). In addition to the solid solutions, in the surface layer, up to 40% of FeMo intermetallides are formed, which contributes to a significant increase in the microhardness of the surface layer after the ESA process at Wp = 0.55 J (at Wp = 0.13 J, Hμ = 5474 MPa, at Wp = 0.55 J, Hμ = 7832 MPa; see Table 2).

The process of sulfomolybdenization of C40 steel by the ESA method at Wp = 3.4 J leads to an increase in the amount of the martensitic phase of up to 30%, as compared with 11% at Wp = 0.55 J, and to the decrease in the amount of residual austenite (FCC phase) from 25% to 6%, and also in the amount of molybdenum disulfide, which reduces to 5%. It should be noted that with increasing discharge energy, the surface roughness increases, which makes it impossible to obtain reliable results of the X-ray diffraction analysis. Therefore, the diffractograms were processed from the surface after cleaning it with 15-μm sandpaper.

When replacing the substrate made of C40 steel with the substrate made of C22 steel, the sulfomolybdenum coating reveals a smaller amount of the FCC phase, namely the residual austenite, at the same discharge energy (see Figure 10). Molybdenum disulfide is even formed at the discharge energy Wp = 0.55 J (3.77%); and at Wp = 3.4 J, about 8% thereof is found on the surface and about 5% is found at a depth down to 15 μm (see Table 4).

## 4. Conclusions

The considerable research on the process of sulfomolybdenizing metal surfaces, which consists of applying a paste containing sulfur onto a steel surface being processed and further electrospark alloying thereof by a molybdenum electrode at discharge energies of 0.13; 0.55; 3.4 J, has led to the following conclusions:Metallographic and durametric studies have shown that the sulfomolybdenum coatings consist of the following four zones:the upper loose layer has the microhardness Hμ of 1112 to 2040 MPa;the “white” strengthened layer has the microhardness Hμ of 5147 to 5474 MPa for Wp = 0.13 J and Hμ of 10596 to 10731 MPa for Wp = 3.4 J;the diffusion zone; andhe base metal.When replacing the substrate of steel C22 with steel C40, there is an increase in the values of the microhardness, the thickness of the strengthened layer, as well as the continuity thereof.Electron microscopic studies of the obtained coatings showed that the resulting layer has a heterogeneous composition with different concentrations of elements. According to the element distribution maps of the areas of the studied specimens, sulfur is concentrated on the surface and molybdenum is more evenly distributed in the coating. The energy dispersion analysis showed that sulfur and molybdenum are concentrated at depths up to 4 and 19 μm, respectively, at Wp = 0.13 J, up to 5 and 25 μm, respectively, at Wp = 0.55 J, and up to 15 and 70 μm, respectively, at Wp = 3.4 J.The X-ray diffraction analysis of the obtained coatings confirms the results of the energy dispersion analysis. At low discharge energies, the phase composition of the coatings on C40 steel is represented by BCC (body centered cubic) solid solution with a lattice period close to ferrite, martensite, FCC (face centered cubic) solid solution, and FeMointermetallide (σ-phase). When replacing the substrate made of C40 steel with the substrate made of C22 steel, the sulfomolybdenum coating reveals a smaller amount of the FCC phase that is residual austenite, at the same discharge energy. Molybdenum disulfide is even formed at the discharge energy Wp = 0.55 J (3.77%); and at Wp = 3.4 J, about 8% thereof is found on the surface, and about 5% thereof is found at a depth down to 15 μm.

## Figures and Tables

**Figure 1 materials-14-06332-f001:**
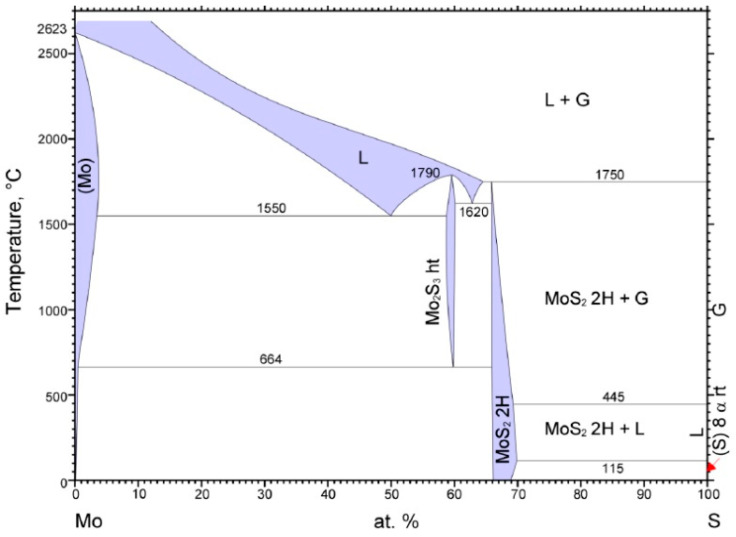
Phase diagram of Mo-S system.

**Figure 2 materials-14-06332-f002:**
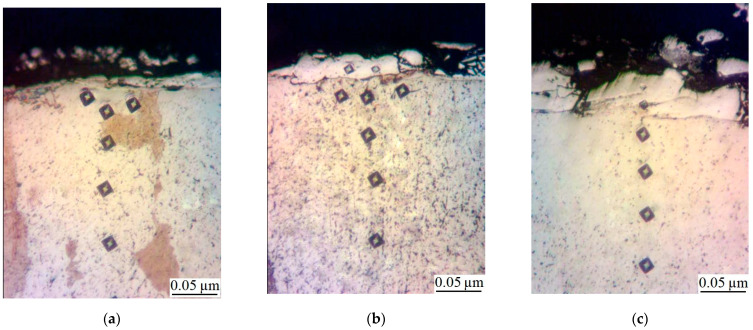
Microstructures in the surface layer of steel C22 after sulfomolybdenizing by ESA method: (**a**) Wp = 0.13 J; (**b**) Wp = 0.55 J; (**c**) Wp = 3.4 J.

**Figure 3 materials-14-06332-f003:**
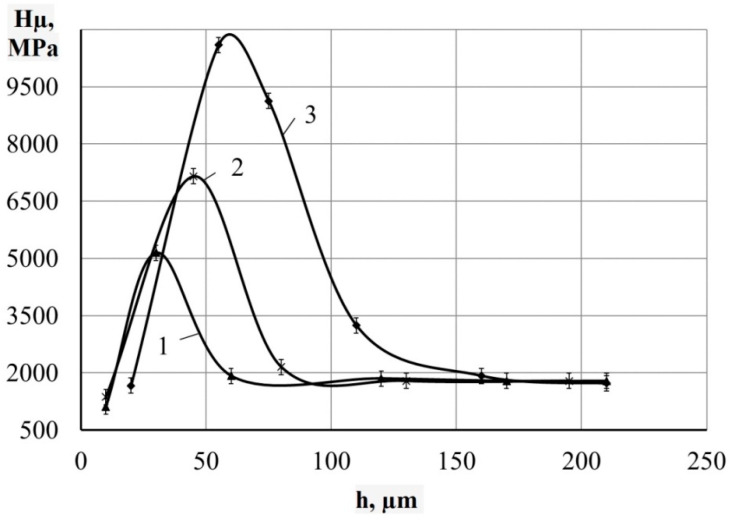
Distribution of microhardness in the surface layer of steel C22 after sulfomolybdenizing by ESA method. On graph: 1—Wp = 0.13 J; 2—Wp = 0.55 J; 3—Wp = 3.4 J.

**Figure 4 materials-14-06332-f004:**
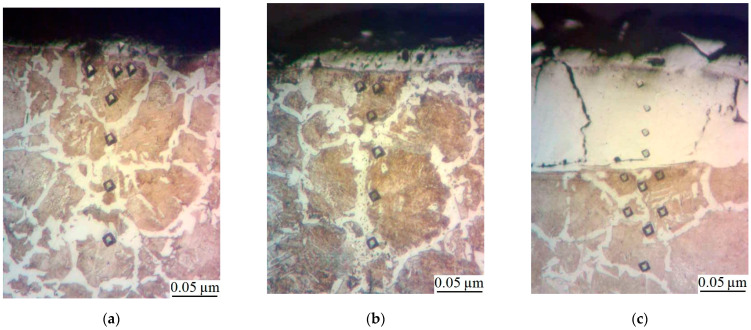
Microstructures in the surface layer of steel C40 after sulfomolybdenizing by ESA method: (**a**) Wp = 0.13 J; (**b**) Wp = 0.55 J; (**c**) Wp = 3.4 J.

**Figure 5 materials-14-06332-f005:**
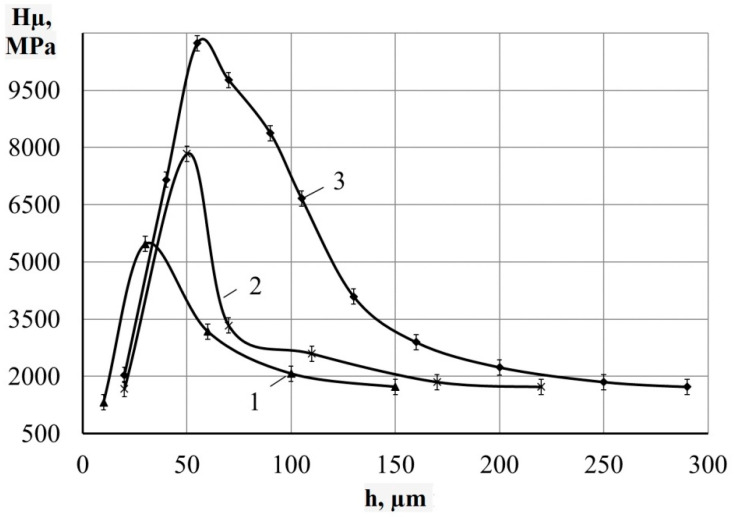
Distribution of microhardness in the surface layer of steel C40 after sulfomolybdenizing by ESA method. On graph: 1—Wp = 0.13 J; 2—Wp = 0.55 J; 3—Wp = 3.4 J.

**Figure 6 materials-14-06332-f006:**
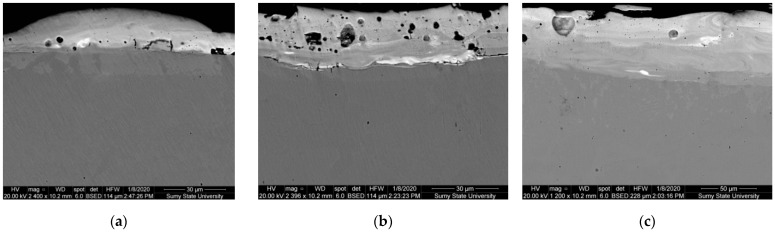
The results of electron microscopic studies of steel C40 sulfomolybdenum coatings obtained by ESA method: (**a**) Wp = 0.13 J; (**b**) Wp = 0.55 J; (**c**) Wp = 3.4 J.

**Figure 7 materials-14-06332-f007:**
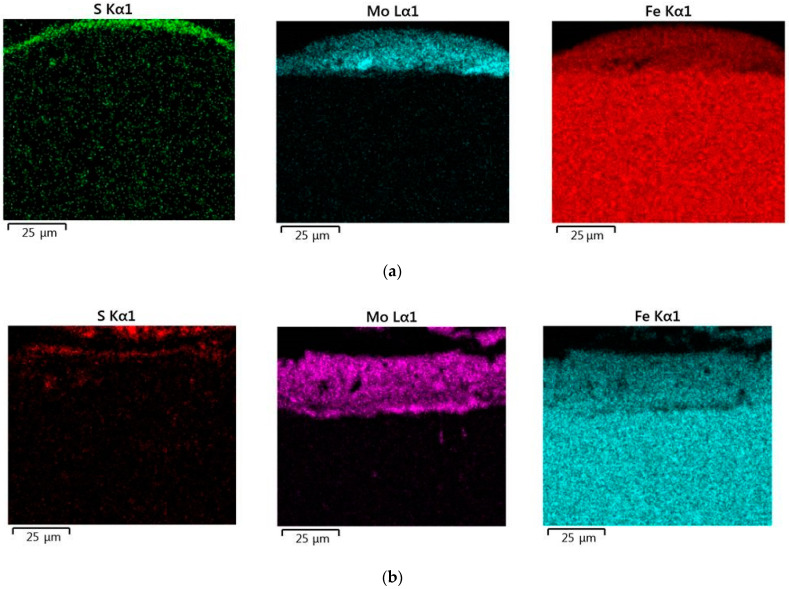
Maps of the distribution of chemical elements in the coating after sulfomolybdenizing steel C40 by ESA method at different discharge energies: (**a**) Wp = 0.13 J; (**b**) Wp = 0.55 J; (**c**) Wp = 3.4 J.

**Figure 8 materials-14-06332-f008:**
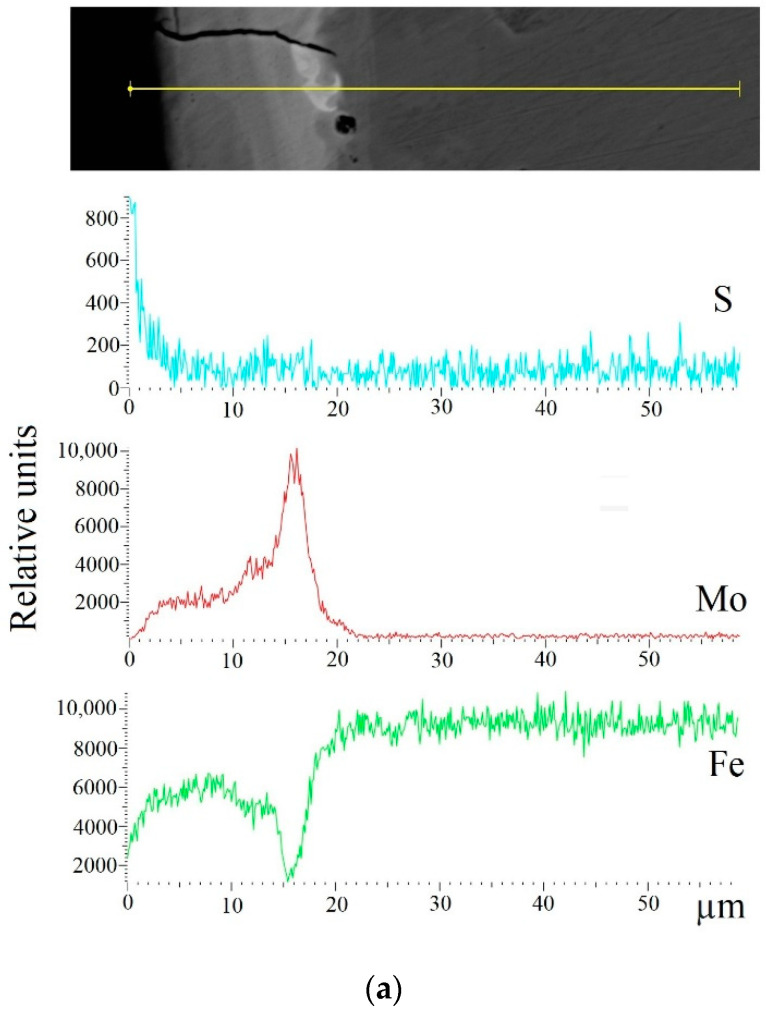
Distribution of elements (sulfur, molybdenum, and iron) in sulfomolybdenum coatings obtained by ESA method on steel C40: (**a**) Wp = 0.13 J; (**b**) Wp = 0.55 J; (**c**) Wp = 3.4 J.

**Figure 9 materials-14-06332-f009:**
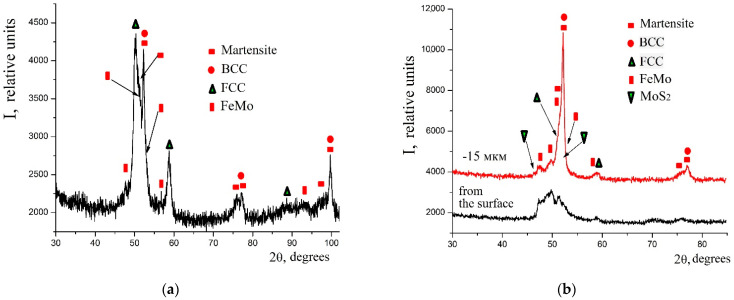
Diffractograms of sulfomolybdenum coatings obtained by ESA method on steel C40: (**a**) Wp = 0.13 J; (**b**) Wp = 3.4 J.

**Figure 10 materials-14-06332-f010:**
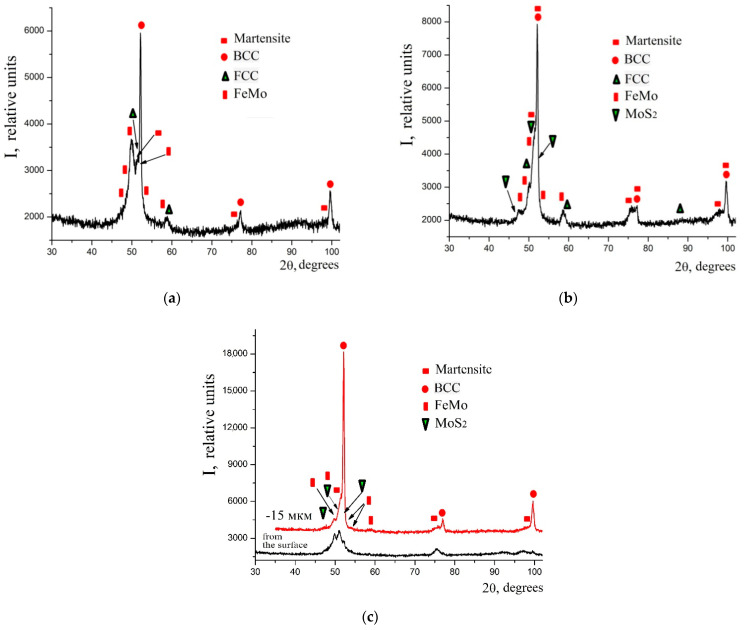
Diffractograms of sulfomolybdenum coatings obtained by ESA method on steel C22: (**a**) Wp = 0.13 J; (**b**) Wp = 0.55 J; (**c**) Wp = 3.4 J.

**Table 1 materials-14-06332-t001:** Dependence of ESA productivity on discharge energy.

Discharge Energy W_p_, J	Productivity, cm^2^/min
0.13	0.8 ÷ 1.0
0.55	1.0 ÷ 1.3
1.3	1.3 ÷ 1.5
2.6	1.5 ÷ 2.0
3.4	2.0 ÷ 2.5
6.8	2.5 ÷ 3.0

**Table 2 materials-14-06332-t002:** Qualitative parameters of sulfomolybdenum coatings obtained by ESA method (S, %—layer continuity).

Discharge Energy, J	Roughness, μm			Layer of Reduced Microhardness			Strengthened Layer		
	Ra	Rz	Rmax	Hµ, MPa	h, μm	S, %	Hµ, MPa	h, μm	S, %
**Steel C22**									
0.13	0.6	2.1	6.1	1112 ± 200	20 ± 5	45 ± 5	5147 ± 200	20 ± 5	65 ± 5
0.55	1.9	3.3	14.2	1368 ± 200	30 ± 5	65 ± 5	7150 ± 200	30 ± 5	75 ± 5
3.40	5.5	14.7	38.5	1666 ± 200	40 ± 5	75 ± 5	10596 ± 200	50 ± 5	90 ± 5
**Steel C40**									
0.13	0.8	2.3	6.5	1320 ± 200	10 ± 5	50 ± 5	5474 ± 200	25 ± 5	75 ± 5
0.55	2.0	3.5	14.7	1670 ± 200	20 ± 5	70 ± 5	7832 ± 200	40 ± 5	90 ± 5
3.40	5.7	14.9	38.7	2040 ± 200	30 ± 5	80 ± 5	10731 ± 200	70 ± 5	95 ± 5

**Table 3 materials-14-06332-t003:** Parameters for crystal lattices of phases and quantitative phase analysis of the sulfomolybdenum coatings on C40 steel.

Discharge Energy, J	Phase	Lattice Period, *a*, nm	Phase Content, % (mass.)
0.55	BCC solid solution	2.8720	23.90
	FCC solid solution	3.6450	25.38
	Martensite	*a* = 2.8740 *c* = 2.9200	10.98
	FeMo (σ-phase)	*a* = 9.1280 *c* = 4.8130	39.74
	At a depth down to 15 µm		
3.4	BCC solid solution	2.8720	46.36
	FCC solid solution	3.6450	6.10
	Martensite	*a* = 2.8640 *c* = 2.9200	30.14
	FeMo (σ-phase)	*a* = 9.1280 *c* = 4.8130	12.53
	MoS_2_	*a* = 3.1212 *c* = 12.2410	4.87

**Table 4 materials-14-06332-t004:** Parameters for crystal lattices of phases and quantitative phase analysis of the sulfomolybdenum coatings on C22 steel.

Discharge Energy, J	Phase	Lattice Period, *a*, nm	Phase Content, % (mass.)
0.13	BCC solid solution	2.8720	39.43
	FCC solid solution	3.6450	16.15
	Martensite	*a* = 2.8640 *c* = 2.9200	18.22
0.55	BCC solid solution	2.8720	34.92
	FCC solid solution	3.6450	13.72
	Martensite	*a* = 2.8640 *c* = 2.9200	30.46
	FeMo (σ-phase)	*a* = 9.1280 *c* = 4.8130	17.14
	MoS_2_	*a* = 3.1212 *c* = 12.2410	3.77
	At a depth down to 15 µm		
3.4	BCC solid solution	2.8800	59.86
	Martensite	*a* = 2.8640 *c* = 2.9200	25.46
	FeMo (σ-phase)	*a* = 9.1280 *c* = 4.8130	9.56
	MoS_2_	*a* = 3.1212 *c* = 12.2410	5.12

## Data Availability

Patent for utility model No 144932, https://base.uipv.org/searchINV/search.php?action=viewdetails&dbname=inv&lang=eng&chapter=biblio&sortby=_ (accessed on 5 October 2021).

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
