# Peer review of "Analysis of the Quality of Sulfomolybdenum Coatings Obtained by Electrospark Alloying Methods"

_materials, 2021, doi:10.3390/ma14216332_

Round 1

Author Response

Thank you for your comments

Reply to Review

«The authors should make clearer the relevance and differences of the work in refs. [25], [26] to their own work and also, make clearer the unique contributions of their method.»

Agree with Remark. We made the necessary additions.

«Figure 9, kindly replace kyrilic words with their English equivalent, for the benefit of the generalreader.»

Agree with Remark. We made the necessary additions.

«Tables 1,2 and 3 use . And not , as decimal point»

Agree with Remark. We made the necessary additions.

Reviewer 2 Report

The paper is focused on the investigation of sulfomolybdenum coatings (obtained by electrospark alloying methods) for steel surfaces.  The topic falls within the scope of the journal. I recommend the publication after the following revision:

  • 6. The scale length within SEM images is not clear. Please check and revise.
  • SEM analyses. The energy of beam and the working distance should be reported in the Experimental section (paragraph 2).
  • Errors for the parameters reported in Table 2 should be added.
  • The quality of Fig. 8 is very poor. Please improve it.
  • Are the wettability properties of the steel samples affected by the sulfomolybdenum coatings? Within this, additional water contact angle experiments could be helpful. Otherwise, the authors should present a hypothesis on the effects of the coatings (obtained at the variable experimental conditions) on the wettability of the steel samples.    

Author Response

Thank you for your comments

Reply to Review

«The scale length within SEM images is not clear. Please check and revise.»

The image has a scale mark, in Figure 6, a, b and c, respectively, 30, 30 and 50 µm. It may be necessary to enlarge the SEM images in the manuscript.

«SEM analyses. The energy of beam and the working distance should be reported in theеexperimental section (paragraph 2).»

Agree with Remark. We made the necessary additions.

«Errors for the parameters reported in Table 2 should be added.»

Agree with Remark. We made the necessary additions.

«The quality of Fig. 8 is very poor. Please improve it.»

Agree with Remark. We made the necessary additions.

«Are the wettability properties of the steel samples affected by the sulfomolybdenum coatings? Within this, additional water contact angle experiments could be helpful. Otherwise, the authors should present a hypothesis on the effects of the coatings (obtained at the variable experimental conditions) on the wettability of the steel samples.»

The effect of wettability sulfomolybdenum coatings of the steel samples has not been investigated. Authors plan to perform wettability tests of obtained coatings in the future as a supplementary properties data. At the current stage of research, the authors do not want to make any hypotheses related to the topic of wettability. 

Round 2

Reviewer 2 Report

The paper can be published in the present form.